# Exploring High-dimensional Search Space via Voronoi Graph Traversing

**Aidong Zhao**[1]      **Xuyang Zhao**[1]      **Tianchen Gu**[1]      **Zhaori Bi**[*,1]      **Xinwei Sun**[2]

**Changhao Yan**[1]          **Fan Yang**[1]          **Dian Zhou**[†,1,3]          **Xuan Zeng**[*,1]

[1]State Key Laboratory of Integrated Chips and Systems, School of Microelectronics, Fudan University, Shanghai, China
[2]School of Data Science, Fudan University, Shanghai, China
[3]Department of Electrical Engineering, University of Texas at Dallas, Richardson, Texas, USA

## Abstract

Bayesian optimization (BO) is a well-established methodology for optimizing costly black-box functions. However, the sparse observations in the high-dimensional search space pose challenges in constructing reliable Gaussian Process (GP) models, which leads to blind exploration of the search space. We propose a novel Voronoi Graph Traversing (VGT) algorithm to extend BO to ultra high-dimensional problems. VGT employs a Voronoi diagram to mesh the design space and transform it into an undirected Voronoi graph. VGT explores the search space by iteratively performing path selection, promising cell sampling, and graph expansion operations. We introduce a UCB-based global traversal strategy to select the path towards promising Voronoi cells. Then we perform local BO within the promising cell and train local GP with a neighboring subset. The intrinsic geometric boundaries and adjacency of the Voronoi graph assist in fine-tuning the trajectory of local BO sampling. We also present a subspace enhancement approach for the intrinsic low-dimensional problems. Experimental results, including both synthetic benchmarks and real-world applications, demonstrate the proposed approach's state-of-the-art performance for tackling ultra high-dimensional problems ranging from hundreds to one thousand dimensions.

## 1 INTRODUCTION

The black-box function optimization is a widespread problem in engineering societies, particularly in domains characterized by computationally expensive or time-consuming evaluations, such as integrated circuit design [Zhao et al., 2023], vehicle design [Yao et al., 2011], and drug discovery [Ou-Yang et al., 2012]. Bayesian optimization (BO), as discussed in [Snoek et al., 2012, Shahriari et al., 2015], is a sample-efficient global optimization for expensive black-box problems. Nonetheless, scaling BO to high-dimensional (HD) space presents significant challenges, and becomes a prominent research area. High-dimensional BO suffers from the hurdles caused by the curse of dimensionality. Firstly, the sparse observations in the HD space compromise the reliability of Gaussian process (GP) models, making it challenging to accurately capture the manifolds of objective functions. Consequently, the imprecise GP models lead to blind exploration across the entire space. Secondly, the computational cost of GP training grows cubically with the number of observations, posing a bottleneck for HD problems. And for the complex HD heterogeneous problems, a large amount of observations is often imperative.

The sample efficiency of BO relies on accurate GP models, which demand numerous observations and become infeasible in HD space. Many high-dimensional BO (HDBO) methods have been proposed to enhance GP reliability and improve the sampling efficiency, as reviewed in Sec.2. Firstly, *dimension decoupling* based methods [Kandasamy et al., 2015, Rolland et al., 2018, Mutný and Krause, 2018, Wang and Jegelka, 2017, Wang et al., 2018] fit the objective function with a set of low-dimensional addictive GPs, avoiding uncertain GP model in the HD space. Secondly, *subspace embedding* based approaches [Wang et al., 2016, Binois et al., 2020, Nayebi et al., 2019, Letham et al., 2020, Eriksson and Jankowiak, 2021, Song et al., 2022] embed the original HD problem into a low-dimensional subspace to obtain an effective subspace GP. However, the presumptions, dimension decomposability or intrinsic low-dimensionality, of these two approaches may not hold in practical problems. Another method, TuRBO [Eriksson et al., 2019], enhances the local reliability of GPs through dynamic trust regions. However, achieving credible local GPs remains computationally expensive and infeasible for problems involving several hundred design variables.

---

[*]Corresponding authors: {zhaori_bi, xzeng}@fudan.edu.cn.
[†]Emeritus Professor, the University of Texas at Dallas.

Local search methods with restart strategies are popular approaches for solving high-dimensional problems, which utilize the *search direction* and *step length* to iteratively navigate towards improved solutions. Commonly used local search methods include line search approaches like the quasi-Newton method BFGS [Nocedal and Wright, 2018, Liu and Nocedal, 1989] and trust region methods such as BOBYQA [Powell, 2009]. However, the global convergence performance of these algorithms significantly relies on the choice of initial solutions.

In this paper, we aim to scale BO to address multi-modal and heterogeneous problems in ultra high-dimensional input spaces. Characterizing function landscapes with regression models within HD spaces spanning several hundred dimensions is inherently impractical. Consequently, we adopt an alternative strategy, steering away from the pursuit of enhancing GP fitness. Inspired by the key idea of local search methods, we leverage the Voronoi boundary and adjacency of each observation $(\boldsymbol{x}, f(\boldsymbol{x}))$ to provide the *step length* and *search direction* information and guide the optimization process. We propose the Voronoi Graph Traversing (VGT) algorithm, which employs Voronoi diagrams to segment the search space into convex Voronoi cells and utilizes adjacency information to construct a Voronoi graph to represent the design space. The contributions of this paper are summarized as follows.

- We propose the VGT algorithm, a sample-efficient approach to enable BO to solve ultra high-dimensional problems. By decomposing the space into Voronoi cells and mapping it to a Voronoi graph, we transform the global exploration in continuous spaces into a Voronoi graph traversal problem. Then the promising cell is identified by traversing the graph via UCB.

- In the local optimization phase, we employ local BO within the selected promising cell. We introduce the Voronoi Neighbored GP (VNGP) model, constructed with the Voronoi neighbors, to reduce computational cost. Additionally, the natural geometric boundaries and Voronoi neighbors assist in fine-tuning the trajectory of local BO sampling.

- For intrinsic low-dimensional problems, we provide a local feature extraction method to capture the local manifold of the objective function and enhance sampling efficiency by targeting effective subspaces.

- We assess the performance of the VGT algorithm using ultra high-dimensional benchmarks, with dimensions extending up to 1000D. The results demonstrate that VGT exhibits exceptional advantages when dealing with high-dimensional problems ranging from hundreds to one thousand dimensions.

A Python implementation of VGT is available on `https://github.com/adzhao072/VGT`.

## 2 RELATED WORKS

The key ideas to tackle HDBO include *dimension decoupling*, *subspace embedding*, and *region restriction*.

*Dimension decoupling* based methods rely on the assumption of dimension decomposability within the objective function. ADD-GP [Kandasamy et al., 2015] is proposed to learn the additive structure and decompose the high-dimensional space into disjoint or overlapping subspaces [Rolland et al., 2018, Mutný and Krause, 2018, Wang and Jegelka, 2017]. However, training a collection of GPs is computationally expensive and unaffordable for large observations. To alleviate the computational cost of GPs, various methods have emerged to approximate the GP kernel with Fourier features [Mutný and Krause, 2018, Rahimi and Recht, 2007, Sriperumbudur and Szabo, 2015, Wang et al., 2018, Hensman et al., 2017]. Nevertheless, the challenge of expensive computation and unknown dimension structure still hinder their application in high-dimensional cases.

*Subspace embedding* is a currently popular method that projects the high-dimensional problem into a low-dimensional subspace based on the assumption of intrinsic low dimensionality. Linear embedding methods, including RemBO [Wang et al., 2016, Binois et al., 2020], HesBO [Nayebi et al., 2019] and ALEBO [Letham et al., 2020], cast the problem into a randomly selected linear subspace and perform BO within the subspace. SAASBO [Eriksson and Jankowiak, 2021] and MCTS-VS [Song et al., 2022] aim to improve BO's sample efficiency by identifying sparse effective variables. Additionally, other methods focus on learning non-linear feature spaces with neural networks [Lu et al., 2018, Tripp et al., 2020, Maus et al., 2022].

*Region restriction* is an effective approach for directly managing the high-dimensional input space. TuRBO [Eriksson et al., 2019] confines the optimization within dynamically adjusted hyper-rectangular trust regions, which resists blind exploration across the entire search space. Extensions of TuRBO have been proposed for categorical and mixed variables [Wan et al., 2021], as well as for faster local descent [Zhai and Gao, 2022]. Another approach, LA-MCTS [Wang et al., 2020, Yang et al., 2021], introduces a SVM-based hierarchical space partition and balances the exploration and exploitation via Monte Carlo tree search (MCTS).

## 3 PROBLEM SETUP AND BACKGROUND

**Problem setup.** We consider the following black-box function optimization problem:

$$\boldsymbol{x}^* = \arg\min_{\boldsymbol{x} \in \mathcal{X}} f(\boldsymbol{x}), \tag{1}$$

where $\boldsymbol{x}$ represents the input variable, $\mathcal{X} = [0, 1]^D$ is the normalized search space, $f : \mathcal{X} \to \mathbb{R}$ denotes the objective function that incurs computationally expensive evaluations,

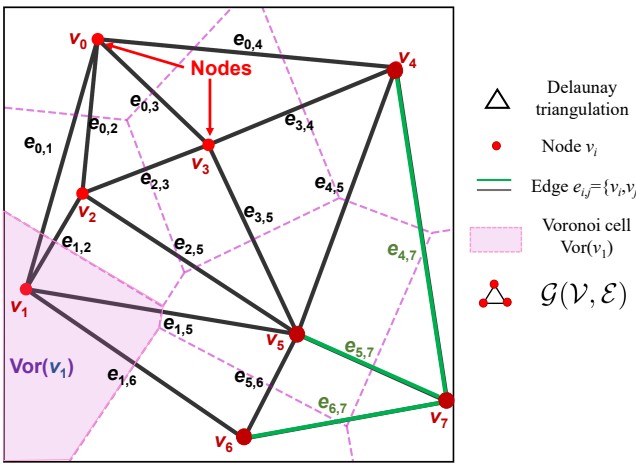

Figure 1: An illustration of a Voronoi graph $\mathcal{G}(\mathcal{V}, \mathcal{E})$. The edge set $\mathcal{E}$ is determined by Delaunay triangulation. The neighbor set corresponding to node $v_7$ is $\mathcal{N}(v_7) = \{v_4, v_5, v_6, v_7\}$.

and $\boldsymbol{x}^*$ is the optimal parameter that achieves the minimal function value.

**Voronoi Diagram.** The Voronoi diagram, also known as Dirichlet tessellation, is a geometric representation that partitions space based on the Euclidean distance to a given set of observations or seeds. Consider a set of $n$ observations denoted as $\mathcal{V} = \{v_i; i = 0, \cdots, n-1\}$ in the space $\mathcal{X}$. The Voronoi diagram divides $\mathcal{X}$ into $n$ convex polygons known as Voronoi cells. An observation $v_j$ serves as the site for its corresponding Voronoi cell $\text{Vor}(v_j)$. The Voronoi cell $\text{Vor}(v_j)$ covers the region that is closer to $v_j$ than any other observation in $\mathcal{V}$ [de Berg et al., 2008]:

$$\text{Vor}(v_j) = \{\boldsymbol{x} \in \mathcal{X} | \forall v_i \in \mathcal{V}, ||\boldsymbol{x} - v_j|| \leq ||\boldsymbol{x} - v_i||\}. \quad (2)$$

The adjacent cells that share common Voronoi boundaries are called Voronoi neighbors. Given an observed dataset $\mathcal{V}$, the Voronoi diagram $\text{Vor}(\mathcal{V})$ is uniquely determined. The neighborhood relationships within the Voronoi diagram can be established through the utilization of Delaunay triangulation. When two Voronoi cells share a common edge, it signifies a neighbor relationship between the corresponding nodes in $\mathcal{X}$. Fig. 1 presents an illustration of the Voronoi diagram along with its dual, the Delaunay triangulation.

**Voronoi Graph.** Based on the Voronoi diagram, we define a Voronoi graph $\mathcal{G}(\mathcal{V}, \mathcal{E})$ as an undirected graph with self-loops. The node set $\mathcal{V}$ consists of the given observations. Each node $v_i$ corresponds to a Voronoi cell $\text{Vor}(v_i)$. Each non-looped edge $\{v_i, v_j\} \in \mathcal{E}$, where $i \neq j$, represents the existence of a specific neighboring relationship between Voronoi cell $\text{Vor}(v_i)$ and $\text{Vor}(v_j)$. If the neighboring relationship is defined by Voronoi adjacency, the Voronoi graph $\mathcal{G}(\mathcal{V}, \mathcal{E})$ extends the Delaunay triangulation by including self-loops. However, in the high-dimensional sce-

narios, the exponential increase in possible simplices, such as triangles or tetrahedrons, presents significant challenges for the efficient construction and representation of Voronoi boundaries and Delaunay triangulation, both in terms of computation and storage. Fortunately, the VGT algorithm does not rely on the exact Voronoi neighboring relationship. For high-dimensional cases, we construct an approximate graph based on the similarity of observations without explicit Voronoi computation. As outlined in Sec.4.3, we introduce a Voronoi graph approximation technique that efficiently captures the neighborhood relationships and builds the connected Voronoi graph. To simplify the notation, we define the neighboring subset centered around a node $v$ as $\mathcal{N}(v) = \{v_i; \text{ where } \{v, v_i\} \in \mathcal{E} \text{ for } v_i \in \mathcal{V}\}$. Notice that, $\mathcal{N}(v) \subseteq \mathcal{V}$ includes both the node $v$ itself and its corresponding neighbors.

**Slice Inverse Regression.** Slice Inverse Regression (SIR) is a supervised method to discover the effective dimension reduction (EDR) directions, particularly in scenarios where there is a limited number of observations in the high-dimensional search space ($n \ll D$). In SIR, a regression model is defined as

$$y = f(\boldsymbol{\beta}_1^T \boldsymbol{x}, \cdots, \boldsymbol{\beta}_d^T \boldsymbol{x}, \epsilon), \quad (3)$$

where $\mathcal{B} = \{\boldsymbol{\beta}_1, \cdots, \boldsymbol{\beta}_d\}$ represents the $d$-dimensional EDR subspace ($d \ll D$), and $\epsilon$ denotes the regression noise. The pattern space $\mathcal{B}$ is extracted from the central inverse regression curve $E(\boldsymbol{x}|y) - E(\boldsymbol{x})$ under the linear design condition (LDC). This is achieved by solving the generalized eigen decomposition problem:

$$\Sigma_{E(\boldsymbol{x}|y)}\boldsymbol{\beta} = \lambda \Sigma_{\boldsymbol{x}}\boldsymbol{\beta}, \quad (4)$$

where $\Sigma_{(\cdot)}$ denotes the covariance matrix empirically estimated using the sliced observations [Li, 1991].

# 4 VORONOI GRAPH TRAVERSING METHOD

In this section, we introduce the Voronoi graph traversing (VGT) algorithm, a novel approach for addressing HD black-box function optimization problems.

## 4.1 VORONOI GRAPH TRAVERSING

Due to limited observations and high computational complexity, building reliable GP models in HD spaces is impractical. Thus, the sample guidance provided by the surrogate model diminishes significantly, resulting in blind exploration of HD space. Our Voronoi Graph Traversing (VGT) algorithm takes a different approach, which employs the geometric information within the Voronoi diagram to implicitly update the *step length* and *search direction*, planning the traversal trajectory in HD space.

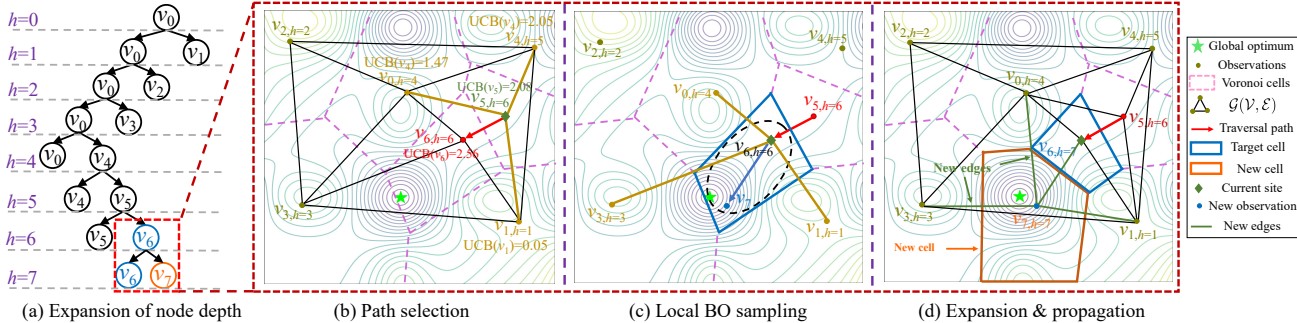

(a) Expansion of node depth | (b) Path selection | (c) Local BO sampling | (d) Expansion & propagation

Figure 2: An illustration of the search procedure of VGT. **(a) Expansion of node depth:** The depth of a node is incremented if a new sample is created within its corresponding Voronoi cell. **(b) Path selection:** Select the red path $v_{5,h=6} \to v_{6,h=6}$ to a promising neighbor via UCB. **(c) Local BO sampling:** Train VNGP model with Voronoi neighbors and create new sample $v_7$ within promising cell $\text{Vor}(v_{6,h=6})$. **(d) Expansion & propagation:** Create a new cell $\text{Vor}(v_{7,h=7})$, and update Voronoi graph. The depth of target node $v_6$ is incremented by 1.

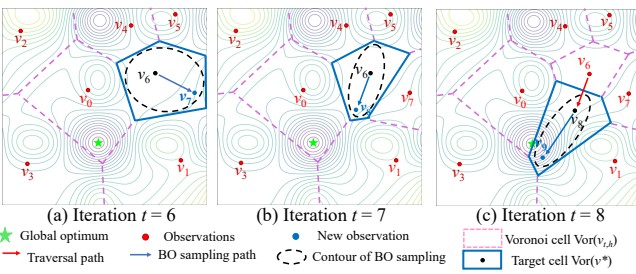

(a) Iteration $t = 6$ | (b) Iteration $t = 7$ | (c) Iteration $t = 8$

★ Global optimum    ● Observations    ● New observation    Voronoi cell $\text{Vor}(v_{i,k})$
→ Traversal path    → BO sampling path    ⬭ Contour of BO sampling    Target cell $\text{Vor}(v^*)$

Figure 3: An illustration demonstrating how Voronoi geometric boundaries guide HD optimization.

We partition the search space into Voronoi cells and represent it as a Voronoi graph utilizing adjacency relationships, as depicted in Fig.1. The global exploration of VGT involves traversing the Voronoi graph, and navigating towards the optimal cell. For the local exploitation phase, the effective selection of *step length* and *search direction* determines the success of the optimization algorithm. We refine the BO sampling mechanism. The Voronoi boundaries, determined by the perpendicular bisectors between a cell and its neighbors, are considered highly informative and deserving of exploration. The GP posterior exhibits a higher standard deviation near these boundaries compared to regions close to existing observations. Sampling around the Voronoi boundaries maximizes the information gain about the current observations $\mathcal{V}$ Contal et al. [2013]. Hence, we anticipate using the Voronoi cell's boundary to guide the selection of the *step length* of the next observation. Additionally, the *search direction* is indicated by the local GP model constructed with Voronoi neighbors of the target cell. However, computing and storing Voronoi boundaries is intractable. We use a Gaussian distribution to approximate the profile of the target Voronoi cell and sample new observations, as indicated by the dashed ellipse in Fig.3. The sample direction of local BO is refined with the assistance of the shape and geometric boundaries of the Voronoi cells. For the "bad" observation with poorer function value, like

$v_7$, the new cell $\text{Vor}(v_7)$ extrudes the space of original cell $\text{Vor}(v_6)$, influencing subsequent sampling directions of BO, as shown in Fig.3(b). Conversely, successful observations, such as $v_8$, guide the algorithm towards more optimal regions, as exemplified by $v_9$ that converges near the global optimum. By leveraging the geometric boundaries and adjacency of the Voronoi graph, VGT precisely utilizes the coordinate and function value of each observation to guide the optimization.

## 4.2 SEARCH PROCEDURE OF VGT

The search procedure is illustrated in Fig.2 and detailed in Algorithm 1, which encompasses three primary stages: (1) *Path selection* creates a movement from the current node towards a promising neighboring node; (2) *Local BO sampling* generates a new sample within the promising Voronoi cell; (3) *Expansion & propagation* expands the graph with new observations and updates the reward. These steps are performed iteratively until the stopping criterion is satisfied.

The traversal algorithm commences by randomly generating an initial node $v_{0,h=0|\mathcal{G}^{t=0}}$, which covers the whole search space $\mathcal{X}$. Here, the node depth $h$ denotes the number of visits to a node and its parents. The update of node depth $h$ follows the tree structure shown in Fig.2(a). The union set of leaf nodes represents a space partition of $\mathcal{X}$ at iteration $t$. Then we present the detailed search procedure of VGT.

**Path Selection.** In each iteration $t$, VGT moves from a start node $v^*_{t-1,h^*_{t-1}|\mathcal{G}^t}$ to a promising neighbor $v^*_{t,h^*_t|\mathcal{G}^t} \in \mathcal{N}(v^*_{t-1,h^*_{t-1}|\mathcal{G}^t})$ (e.g., from node $v_{5,h=6}$ to node $v_{6,h=6}$ in Fig.2(b)), facilitating progress towards the global optimum. Notably, the start node $v^*_{t-1,h^*_{t-1}|\mathcal{G}^t}$ is the optimal node selected in iteration $t - 1$, and also included in $\mathcal{N}(v^*_{t-1,h^*_{t-1}|\mathcal{G}^t})$, allowing for a "stationary step". To achieve a comprehensive exploration of the search space,

**Algorithm 1** Voronoi Graph Traversing (VGT)

---

**Input:** Objective function $f(\boldsymbol{x})$, search space $\mathcal{X}$, maximal iteration $T$.

1: Randomly sample the initial node $\mathcal{V} = \{v_{0,h=0} = \text{random}(\boldsymbol{x}_0, f(\boldsymbol{x}_0))\}$; $\mathcal{G}^{t=0}(\mathcal{V}, \mathcal{E})$.  $\triangleright$ Random Initialization
2: Traverse from $v^*_{-1,h^*_{-1}|\mathcal{G}^{t=0}} \leftarrow v_{0,h=0|\mathcal{G}^{t=0}}$.
3: **for** $t = 0$ to $T - 1$ **do**
4:     Select the promising node from neighboring subset $v^*_{t,h^*_t|\mathcal{G}^t} = \underset{v_j \in \mathcal{N}(v^*_{t-1,h^*_{t-1}|\mathcal{G}^t})}{\arg\max} \text{UCB}(v_j)$.  $\triangleright$ Path Selection
5:     Train the VNGP model with dataset $\mathcal{N}(v^*_{t,h^*_t|\mathcal{G}^t})$.
6:     Generate sample $\boldsymbol{x}^{t+1}$ by performing BO within Voronoi cell $\text{Vor}(v^*_{t,h^*_t|\mathcal{G}^t})$.  $\triangleright$ Sample via Local BO
7:     Evaluate objective function and collect samples $v_{t+1,h^*_t+1} = (\boldsymbol{x}_{t+1}, f(\boldsymbol{x}_{t+1}))$.
8:     $\mathcal{G}^{t+1} \leftarrow \mathcal{G}^t.\text{append}(v_{t+1,h^*_t+1})$; $v^*_{t,h^*_t+1|\mathcal{G}^{t+1}} \leftarrow v^*_{t,h^*_t+1|\mathcal{G}^t}$; Update UCB.  $\triangleright$ Expansion & Propagation
9: **end for**

---

we employ the Upper Confidence Bound (UCB) criterion to assess the potential of each node during the dynamic path selection process. The UCB for VGT is defined as follows [Lupu and Precup, 2018]:

$$\text{UCB}(v_i) = Q(v_i) + \sqrt{\frac{C_p \cdot \ln t}{h(v_i)}}, \quad (5)$$

where $Q(v_i)$ represents the quality of node $v_i$, and $h(v_i)$ denotes the depth of $v_i$. For simplicity, we estimate the quality of node $v_i$ as $Q(v_i) = -f(\boldsymbol{x}_i)$. The hyperparameter $C_p$ balances the exploration of under-explored areas and the exploitation of promising regions. The traversal algorithm in VGT specifically limits the evaluations of UCB to nodes that have established neighborhood relationships, rather than evaluating all nodes in the graph. This selective UCB evaluation strategy aims to prioritize nodes that are more likely to contribute to finding the optimal solution. By focusing on nodes with established neighborhood relationships, VGT ensures a more targeted exploration.

**Sample via Local BO.** Once a promising node is selected, we perform local BO within the corresponding Voronoi cell. To mitigate the computational complexity associated with GP modeling, we propose the Voronoi Neighbored GP (VNGP) model, which is trained with the neighbor dataset $\mathcal{N}(v^*_{t,h^*_t|\mathcal{G}^t})$. As the GP kernel is correlated with the distance between observations, the observations located far from the target cell contribute little to local modeling. VNGP takes advantage of the spatial structure of the Voronoi graph to enable computationally efficient local modeling while maintaining accuracy. The new observation $v_{t+1,h^*_t+1}$ is sampled by optimizing the acquisition function within the irregular Voronoi cell (e.g., the blue polygonal region in Fig.2(c)). To achieve this, we employ a Gaussian distribution (e.g., the black dashed ellipse in Fig.2(c)) centered around the Voronoi site $v^*_{t,h^*_t|\mathcal{G}^t}$ to sample the acquisition function. The hyperparameters of the sample distribution are tuned using random samples located within the target cell. Samples lying outside the target cell are discarded by reject sampling.

**Expansion & Propagation.** Each sample within a target cell $\text{Vor}(v^*_{t,h^*_t|\mathcal{G}^t})$ of depth $h^*_t$ creates a new cell of depth $h^*_t + 1$ while also incrementing the depth of $v^*_{t,h^*_t|\mathcal{G}^{t+1}}$. Once a new observation is sampled, the graph $\mathcal{G}^{t+1}$ is then expanded by incorporating a new Voronoi cell centered around the new observation $v_{t+1,h^*_t+1}$, as shown in Fig.2(d). The UCB is updated to refine the trajectory of the next step based on the most recent information. The *expansion & propagation* step progressively expands the coverage of search space by incorporating the newly acquired nodes and enhances the algorithm's global exploration capability.

By iteratively performing the aforementioned steps, the algorithm continues to explore the design space and adjust its trajectory to efficiently navigate towards the optimal region. The VNGP model is dynamically updated along with the movement and incorporation of new samples to maintain its local responsiveness and adaptability. By leveraging the VNGP model and employing the UCB selection strategy, the algorithm identifies a traversal path that progresses towards the global optimum along the valley bottom, as illustrated by the red traversing path in Fig.4. For a visual overview of the VGT traversal procedure, please refer to Appendix A.

### 4.3 SCALING TO HIGH-DIMENSIONAL SEARCH SPACE

In this section, we propose two key strategies, namely Voronoi graph approximation and subspace BO sampling, to tackle the challenges of scaling the VGT algorithm to high-dimensional search spaces.

**Voronoi Graph Approximation.** Determining the Voronoi diagram and Delaunay triangulation in high-dimensional space and large-sample-budget scenarios is computationally infeasible. Instead of explicitly calculating the Voronoi boundaries and Delaunay connections, we can employ similarity search approaches, such as $K$-nearest neighbor search (K-NNS) [Yianilos, 1993] or approximate nearest neighbor search (ANNS) [Andoni

et al., 2015, Malkov and Yashunin, 2020], to discover the neighborhood relationships among observations and construct an approximate Voronoi graph. In this work, we utilize K-NNS to approximate the neighborhood of a given node. By identifying the $K$ nearest neighbor nodes, we can establish connections between each node and its $K$ closest neighbors, thereby forming a connected graph within the search space. $K$ is a hyper-parameter that depends on the dimensionality of the problem. With a larger $K$, the Voronoi neighbors will be included in the $K$ nearest neighbors. For a moderate $K$, the Voronoi neighbors and the $K$ nearest neighbors often coincide. Additionally, in high-dimensional cases, the Voronoi boundaries are not computed explicitly. Instead, we use reject sampling to discard candidates outside the target cell based on the property described in Eq.(2). Ultimately, the high-dimensional Voronoi graph approximation problem boils down to K-NNS. This approximation effectively captures the local neighborhood relationships while avoiding the computational overhead associated with explicit Voronoi boundaries and Delaunay connections.

**Subspace BO Sampling.** In high-dimensional spaces, the localized GP model is often underfitting and exhibits large uncertainty, especially when the number of available samples is significantly smaller than the problem's dimension $D$. To tackle this issue, we incorporate Localized SIR (LSIR) [Wu et al., 2008] to capture the local EDR subspace denoted as $\mathcal{B}^t$ by leveraging information from neighboring samples. Furthermore, experience suggests that the objective function tends to decrease along the previous descent direction denoted as $s^{t-1}$. Exploiting this insight, we construct the pattern subspace $\mathcal{S}^t = \{s^{t-1}\} \cup \mathcal{B}^t$ with a dimension significantly smaller than the original problem's dimension, denoted as $|\mathcal{S}^t| \ll D$. Consequently, by accurately modeling the subspace problem with a smaller number of observations, we can effectively optimize the acquisition function $\alpha(x^t + s)$ within the subspace $\mathcal{S}^t$, subject to the step length constraint $x^t + s \in \text{Vor}(v^*_{t,h^*_t|\mathcal{G}^t})$:

$$
\begin{aligned}
s^t &= \arg\max_{s \in \mathcal{S}^t} \alpha(x^t + s), \\
\text{s.t. } &\ x^t + s \in \text{Vor}(v^*_{t,h^*_t|\mathcal{G}^t}),
\end{aligned} \tag{6}
$$

where $\alpha(\cdot)$ is the acquisition function. The subspace method can efficiently capture the local effective manifold of objective functions in ultra HD space and enhance local BO sampling. To mitigate the potential degradation of diversity caused by subspace BO, we adopt a strategy of alternating subspace sampling and full-dimension sampling. We introduce a hyper-parameter $R_p$ to represent the ratio between subspace sampling and full-dimension sampling. For sparse optimization problems, where the valid dimensions are limited, a larger value of $R_p$ can be chosen to allocate more iterations for exploiting the subspace spanned by the effective feature directions. On the other hand, for dense

problems with a larger number of relevant dimensions, a smaller value of $R_p$ can be utilized to focus more on the exploration of the promising cells.

# 5 DISCUSSIONS

## 5.1 COMPLEXITY ANALYSIS

The computational burden of the VGT algorithm mainly originates from two factors: training the VNGP model and optimizing the acquisition function. For each iteration $t$, fitting the VNGP model incurs a complexity of $\mathcal{O}(K^3)$. Optimizing the acquisition function involves the prediction complexity of the VNGP model, which is $\mathcal{O}(K^2)$, as well as the complexity of the nearest neighbor search (NNS). The complexity of NNS depends on the specific implementation. In our approach, we utilize the popular *k-d* tree for NNS, which involves depth-first tree traversal and backtracking. The backtracking operation typically grows exponentially with the dimension $D$, resulting in a linear query complexity for high-dimensional problems. In many scenarios, the *k-d* tree struggles to outperform brute-force search , which has a search complexity of $\mathcal{O}(D \cdot N)$, due to the curse of dimensionality. This work is primarily focused on enhancing the sample efficiency of HDBO, and we do not delve into the challenges associated with high-dimensional NNS further. Initially, the computational cost is predominantly dominated by the VNGP model, while in later iterations, NNS becomes the main factor affecting computational efficiency. For a more detailed description of the computational complexity of VGT, please refer to Appendix B.

## 5.2 INSIGHTS

In this paper, we propose VGT as an efficient global optimization approach for complex and heterogeneous problems over HD search space. VGT divides the design space into Voronoi cells and traverses the graph to achieve global exploration. By combining graph traversal with promising cell sampling, VGT guides towards the optimal region along a valley with small function values, as shown in Fig.4. Regions with poor function values are effectively avoided with a few additional observations. While VGT operates within the localized BO framework, maintaining a global perspective is crucial for ensuring the quality of convergence. In addition to the graph traversal strategy, VGT employs a restart mechanism to enhance its global search capability. If there is no reduction in the objective function value over several consecutive iterations, the algorithm moves to the Voronoi cell with the minimal depth $h$ and creates a new search path.

The Voronoi diagram in VGT provides a fine-grained partition of the search space, with each observation contributing to the update of the geometry of the promising region and

guiding the search direction. Compared to TuRBO, which employs hyper-rectangular trust regions, VGT leverages the geometric boundaries defined by each observation, and adapts more effectively to irregular, multi-modal, and heterogeneous function landscapes. Compared to La-MCTS, which utilizes SVM for domain decomposition, VGT exhibits lower computational complexity and superior scalability to high-dimensional heterogeneous problems, as the SVM boundary inherently relies on the adaptation of the kernel function to the objective function landscape.

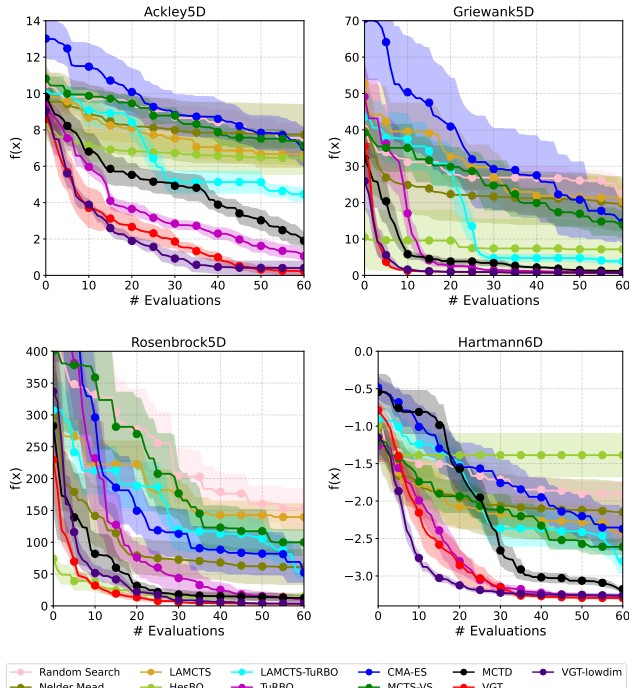

Figure 5: Optimization results for low-dimensional synthetic benchmarks.

parameters, and additional experimental results, please refer to Appendix C.

## 6.1 SYNTHETIC FUNCTIONS

### 6.1.1 Low-dimensional Synthetic Benchmarks

We first provide a set of low-dimensional synthetic benchmarks to evaluate the performance of the proposed VGT algorithm, including Ackley, Griewank, Rosenbrock, and Hartmann6. The experimental results are presented in Fig.5. For these small-scale problems, we can explicitly compute the Voronoi boundaries and neighbors. Then, we compare the performance of `VGT-lowdim` with explicit Voronoi boundaries and neighbors, and `VGT` with approximated Voronoi graph by K-NNS. Experimental results indicate that the Voronoi graph approximation approach proposed in Sec.4.3 does not compromise VGT's sample efficiency. Both of the aforementioned VGT methods achieve superior sampling efficiency and better solutions compared to state-of-the-art baselines, with TuRBO and MCTD following behind. The performance of the subspace embedding-based approaches, HesBO and MCTS-VS, is inferior to the local BO methods.

### 6.1.2 High-dimensional Synthetic Benchmarks

We evaluate the performance of VGT and compare it against various baselines on benchmark functions including Ackley, and Griewank. Both functions are evaluated in dimensions

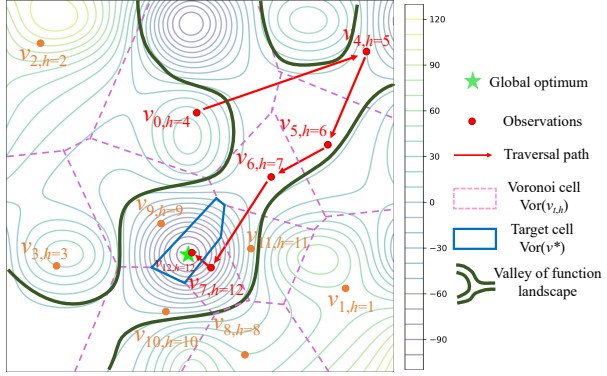

Figure 4: An illustration of the traversal path of VGT. Most observations are concentrated in the promising valley. The traversal path is marked by red arrows.

## 6 EXPERIMENTAL RESULTS

We conduct a thorough evaluation of VGT's performance using a diverse set of HD experiments. Our experiments encompass both synthetic functions, such as Ackley, Griewank, Rosenbrock and Hartmann6, as well as real-world applications, including vehicle design (124D Mopta08), machine learning tasks (388D SVM training), and analog circuit optimization (36D opamp, 77D phase lock loop). Except for the experiments in Sec.6.1.3 involving additional dummy dimensions, all other benchmarks are based on real dimensions and challenging for optimization algorithms.

To provide a comprehensive evaluation, we compare the performance of VGT against a wide range of state-of-the-art baselines, including the local BO methods MCTD [Zhai and Gao, 2022], TuRBO [Eriksson et al., 2019] and La-MCTS [Wang et al., 2020], the subspace embedding-based approaches MCTS-VS [Song et al., 2022] and HesBO [Nayebi et al., 2019], the popular evolutionary algorithm CMA-ES [Hansen et al., 2003], the simplex method Nelder-Mead [Nelder and Mead, 1965], and Random Search. All experiments are conducted on a Linux workstation equipped with Intel Xeon Gold 6230 @2.1GHz CPUs and 128GB memory. To account for random variations, each experiment is repeated 10 times with different random seeds. For more detailed experimental settings, sensitivity analysis of the hyper-

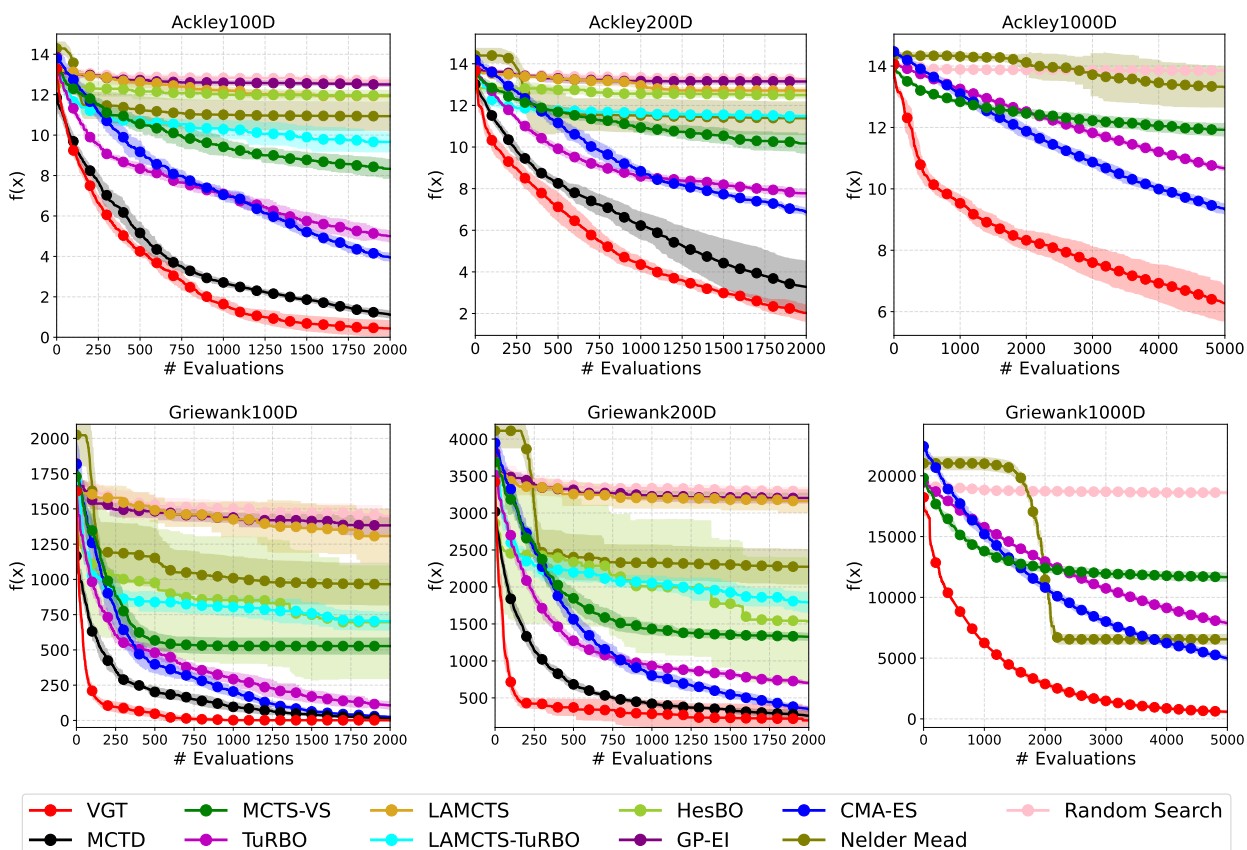

Figure 6: Optimization results for high-dimensional synthetic benchmarks are presented. All benchmarks are multi-modal and challenging for global optimization algorithms, with each having a unique global optimum of 0.

of 100D, 200D and 1000D, and are challenging for global optimization algorithms.

Fig.6 gives a visual comparison of VGT against the baselines. In the 100D and 200D scenarios, VGT consistently outperforms MCTD for both benchmarks, with CMA-ES and TuRBO closely following. The subspace embedding-based method HesBO and variable selection-based method MCTS-VS show unsatisfactory performance when applied to the full-dimensional problems.

For the 1000D ultra high-dimensional scenarios, MCTD encounters difficulties when applying its local descent strategy, resulting in premature termination. The curve of LA-MCTS is also missing due to the computation time exceeding 200 hours, highlighting the computational challenges of the problem. Among the compared algorithms, only VGT demonstrates efficient and stable descent for the ultra high-dimensional cases. CMA-ES and other HDBO methods, including MCTS-VS and TuRBO, are lagging far behind VGT. These results demonstrate that VGT is a highly effective algorithm for ultra HD optimization problems, outperforming other state-of-the-art methods in terms of both sample efficiency and quality of solutions.

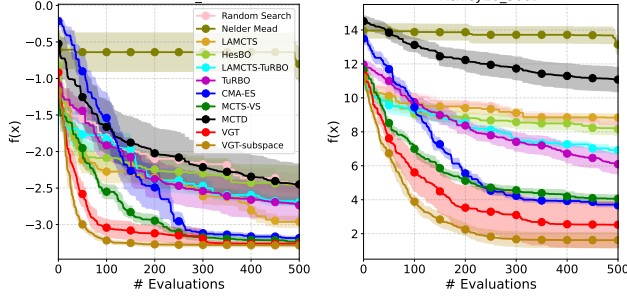

Figure 7: Experiments with additional dummy dimensions.

### 6.1.3 Scenarios with Additional Dummy Dimensions

To evaluate the effectiveness of the proposed subspace sampling method, we conduct experiments with additional dummy dimensions in the Hartmann6D and Ackley10D functions. By extending the dimensions to 500D through the addition of independent dummy variables, we investigate two scenarios of VGT: single full-dimensional sampling `VGT` and effective subspace sampling `VGT-subspace`. For the subspace sampling, we set $R_p = 1/1$ to enhance the subspace exploitation while also maintaining global search capability. The numerical results, depicted in Fig.7, demon-

strate that `VGT-subspace` with subspace sampling exhibited the fastest descent rate among the compared algorithms, closely followed by `VGT` with full-dimensional sampling. The variable selection-based MCTS-VS shows the ability to capture the valid dimensions and achieves satisfactory results. CMA-ES also demonstrates good adaptability to these problems. However, MCTD falls behind TuRBO and is not well-suited for these intrinsically low-dimensional problems. These findings validate the effectiveness of the subspace sampling method in VGT, which allows for sample-efficient optimization even in the presence of additional dummy dimensions.

## 6.2 REAL-WORLD APPLICATIONS

For the real-world optimization problems, we focus on two analog integrated circuit optimization problems based on the open-source benchmark circuits [Sunter and Sarson, 2017], a 36D opamp circuit and a 77D phase lock loop (PLL). Additionally, we consider a 124D soft-constrained vehicle design problem MOPTA08, as well as a 388D SVM training task. For these real-world problems with unknown dimensional structures, we use the parameter setting $R_p = 1/4$ for subspace sampling, which allows us to explore potential EDR directions with a small number of observations.

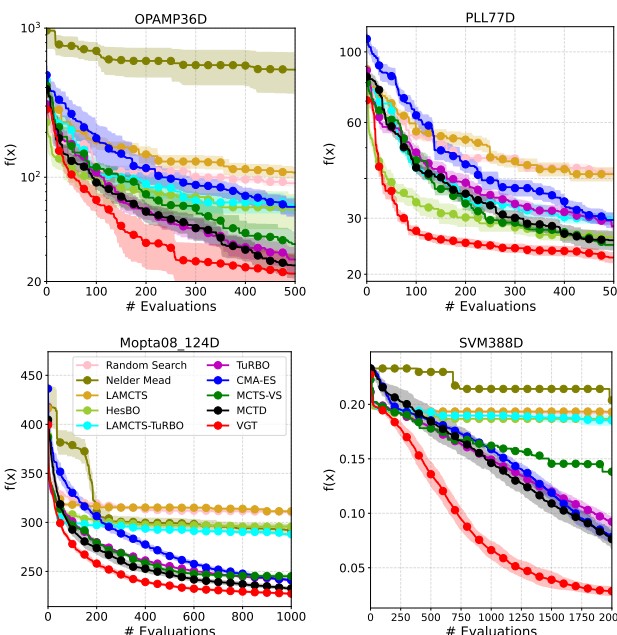

Figure 8: Optimization results for real-world applications. Each practical problem is transformed into a scalar minimization problem.

For the opamp circuit, the objective is to minimize the `Iddq` with three specification constraints. We formulate a scalar objective function with soft penalties to address the circuit design, which involves 36 free parameters related to transistor sizes and capacitor areas. The circuit performance

is obtained from the SPICE simulator, and the objective function exhibits heterogeneity due to the piece-wise device model. VGT outperforms other methods in this case. MCTD achieves a similar descent speed to TuRBO, indicating that the stochastic three-point descent of MCTD can not provide an advantage in this scenario. The simplex method, Nelder Mead, fails to find the feasible region. For the PLL circuit, the objective is to minimize the average current consumption (`Iddavg`) while maintaining the output peak-to-peak voltage. We focus on optimizing the charge pump and voltage-controlled oscillator (VCO) components. Behavior models of logic gates are used to reduce simulation time. The PLL has 77 design parameters related to device sizes. Results show that VGT still outperforms other methods in this case, with HesBO following.

For the 124D soft-constrained vehicle design problem MOPTA08 and the 388D SVM training task, VGT maintains its superiority over other methods in these benchmarks. For MOPTA08, VGT converges to the optimum with a small number of samples, while MCTD, TuRBO, MCTS-VS, and CMA-ES achieve similar final results. In the case of SVM388D, VGT outperforms other baselines by a large margin, highlighting its superiority for high-dimensional problems. MCTD, TuRBO, and CMA-ES also display good performance in high-dimensional settings, while other methods fail to find reasonable solutions. Please refer to Fig.8 for a visual representation of the results.

## 7 CONCLUSIONS

We propose a novel Voronoi graph traversing method for scaling BO to ultra high-dimensional input space. We utilize a UCB-based graph traversing strategy to navigate the search direction in high-dimensional space. Local exploitation efficiency is ensured by sampling within the promising Voronoi cell. Moreover, we provide an efficient subspace BO sampling by restricting BO to the effective subspace extracted using LSIR. Experiments on the ultra high-dimensional benchmarks spanning up to 1000D demonstrate the remarkable advantages of the VGT algorithm for solving problems in ultra high-dimensional input space. The extension of VGT to ultra high-dimensional constrained optimization, multi-objective optimization, and distributed parallel computing is a focus of future research.

**Acknowledgements**

This research is supported partly by the National Natural Science Foundation of China (NSFC) research projects 62141407, 62304052, 92373207, and 12331009.

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

# A CONVERGENCE PROCEDURE OF VGT

In this section, we present a detailed illustration of the iterative and convergence process of the VGT algorithm, depicted in Figure 9, to facilitate readers' in-depth comprehension of VGT's search mechanism in high-dimensional spaces.

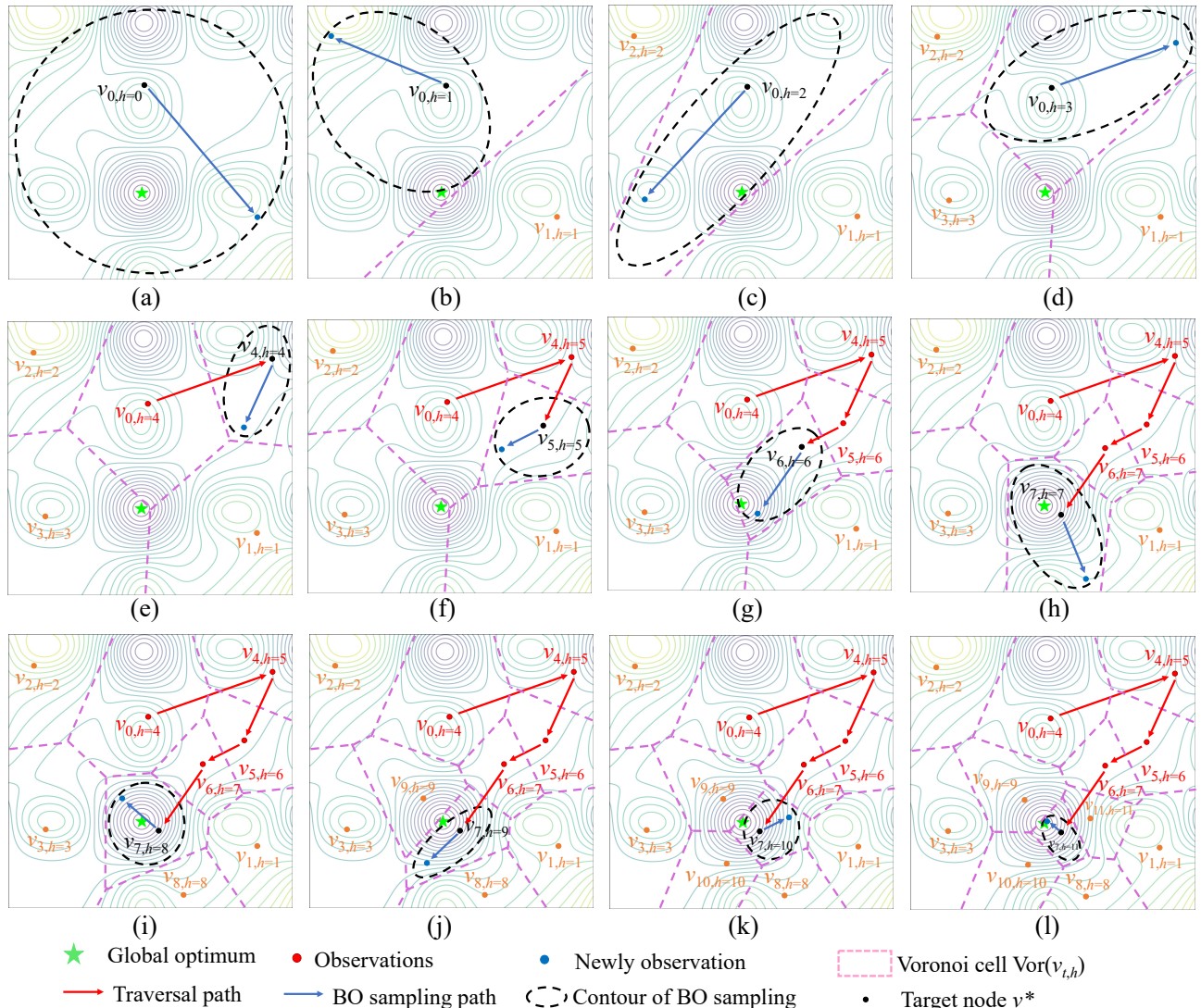

Figure 9: An illustration of the convergence procedure in VGT.

In high-dimensional spaces, limited observations result in highly imprecise GP models, rendering BO prone to blind exploration. The VGT algorithm introduces Voronoi domain decomposition to partition the high-dimensional space into convex Voronoi cells. Global exploration of space is achieved through traversal along the Voronoi graph. Subsequently, the Voronoi cell's geometric boundaries are employed to direct the sampling for local BO. To guarantee the global coverage of the entire space, in each iteration, we aim to position the sampling for local BO as near as possible to the edge of the promising Voronoi cell, typically around the vicinity of the dashed ellipse in Figure 9.

The algorithm starts with a randomly initial point, e.g. $v_0$ as depicted in Figure 9(a), representing the entire search space $\mathcal{X}$. By iteratively performing path selection, promising cell sampling, and graph expansion operations, new cells are generated. The "good" observations, (e.g. $v_4$ in Figure 9(e)), direct the algorithm towards more promising regions (indicated by the red arrow). Conversely, the "bad" observations, like $v_1$, $v_2$, $v_3$, and $v_8$, guide the algorithm away from under-performing areas. With the progression of iterations, the promising cell gradually contracts and converges towards the global optimum.

# B COMPUTATIONAL COMPLEXITY ANALYSIS

The computational burden of the VGT algorithm mainly originates from two factors: training the VNGP model and optimizing the acquisition function. The training and prediction complexity of the VNGP model is influenced by the number of neighbors of the selected node. In the Voronoi graph constructed with $K$ nearest neighbor search (NNS), the promising node is connected to its $K$ nearest neighbors. Consequently, the data size for training the VNGP model is $K$.

We consider a VNGP model with a training dataset $(X_K, \boldsymbol{y}_K)$ of size $K$. The predictive mean $\mu(\boldsymbol{x})$ and variance $\sigma^2(\boldsymbol{x})$ at $\boldsymbol{x} \in \mathcal{X}$ is:

$$\begin{aligned}
\mu(\boldsymbol{x}) &= m(\boldsymbol{x}) + \boldsymbol{k}_K^{\mathrm{T}} C_K^{-1}(\boldsymbol{y} - m(X)), \\
\sigma^2(\boldsymbol{x}) &= k(\boldsymbol{x}, \boldsymbol{x}) - \boldsymbol{k}_K^{\mathrm{T}} C_K^{-1} \boldsymbol{k}_K,
\end{aligned} \tag{7}$$

where $\boldsymbol{k}_K = [k(\boldsymbol{x}, \boldsymbol{x}_0), \cdots, k(\boldsymbol{x}, \boldsymbol{x}_{K-1})]^{\mathrm{T}}$ and $C_K$ is the $K$-dimensional covariance matrix. The model parameters are tuned by maximizing the log marginal likelihood function:

$$L(\theta) = -\frac{1}{2}\boldsymbol{y}^{\mathrm{T}} C_K^{-1} \boldsymbol{y} - \frac{1}{2}\log|C_K| - \frac{K}{2}\log(2\pi). \tag{8}$$

Hence, training the VNGP model has a complexity of $\mathcal{O}(K^3)$ due to the covariance matrix inversion in Eq.(8). The prediction complexity is $\mathcal{O}(K^2)$ in Eq.(7).

Another important computation cost arises from the optimization of the acquisition function within the Voronoi cell for each iteration $t$, which involves the prediction complexity of the VNGP model $\mathcal{O}(K^2)$, and the NNS complexity by $k$-$d$ tree. The $k$-$d$ tree has a construction complexity of $\mathcal{O}(D \cdot N \cdot \log(N))$ and a worst-case query complexity of $\mathcal{O}(D \cdot N)$.

# C EXPERIMENTS

## C.1 RUNTIME COMPARISON

Table 1 presents a comparison of the experimental results and the average runtime for a total of 2000 function evaluations.

Table 1: Runtime comparison of Bayesian optimization methods

| Algorithms | Ackley100D | | Ackley200D | | Griewank100D | | Griewank200D | |
|---|---|---|---|---|---|---|---|---|
| | $f(\boldsymbol{x})$ | Runtime | $f(\boldsymbol{x})$ | Runtime | $f(\boldsymbol{x})$ | Runtime | $f(\boldsymbol{x})$ | Runtime |
| VGT | **0.44 ± 0.42** | **1.0h** | **2.03 ± 0.42** | 1.6h | **0.90 ± 0.13** | 2.2h | **195.6 ± 194.4** | 2.2h |
| MCTD | 1.12 ± 0.20 | 14.1h | 3.28 ± 1.26 | 25.6h | 13.1 ± 6.6 | 4.8h | 259.4 ± 65.1 | 4.5h |
| MCTS-VS | 8.33 ± 0.50 | 1.2h | 10.17 ± 0.49 | **1.2h** | 527.0 ± 60.9 | **1.2h** | 1327.0 ± 55.4 | **47min** |
| TuRBO | 5.01 ± 0.30 | 1.5h | 7.78 ± 0.22 | 1.4h | 107.3 ± 30.6 | 1.6h | 703.7 ± 26.7 | 53min |
| LaMCTS | 11.93 ± 0.12 | 5.3h | 12.72 ± 0.15 | 10.9h | 1307.0 ± 192.5 | 4.9h | 3162.5 ± 163.8 | 6.6h |
| LaMCTS-TuRBO | 9.66 ± 0.52 | 3.7h | 11.48 ± 0.11 | 2.6h | 703.9 ± 68.0 | 17.7h | 1789.3 ± 151.5 | 7.5h |
| GP-EI | 12.50 ± 0.13 | 1.6h | 13.15 ± 0.16 | 3.9h | 1382.9 ± 68.6 | 3.1h | 3203.1 ± 75.0 | 3.7h |

## C.2 ADDITIONAL EXPERIMENTAL RESULTS

For the additional experiments, we consider the valley-shaped function Rosenbrock and the multi-modal function Rastrigin. Both functions are evaluated in dimensions of 100D, 200D, and 1000D, posing challenges for global optimization algorithms.

Fig. 10 presents the experimental results of 10 repeated runs. For the 100D and 200D Rosenbrock function, VGT consistently outperforms MCTD, with CMA-ES and TuRBO closely following behind. CMA-ES and TuRBO exhibit comparable sample efficiency for this case. The subspace embedding-based method HesBO and variable selection-based method MCTS-VS show unsatisfactory performance when applied to full-dimensional problems.

For the 100D and 200D Rastrigin benchmarks, VGT initially lags behind MCTD due to its strategy of investing more observations in exploring the search space and assessing potential regions. However, VGT eventually surpasses them and achieves the best final solutions.

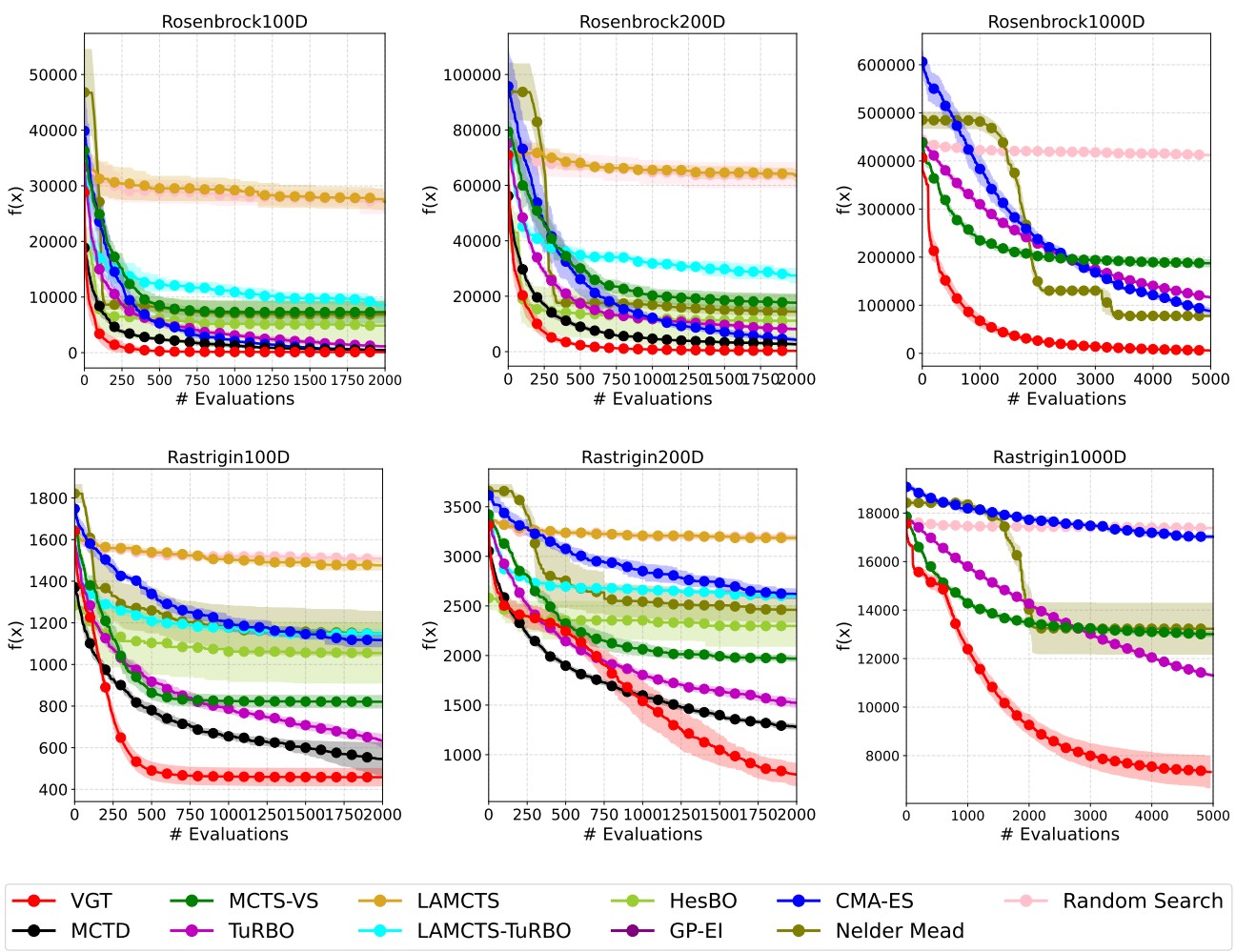

Figure 10: Optimization results for high-dimensional synthetic benchmarks.

For the 1000D ultra high-dimensional scenarios, MCTD still fails to handle the ultra high-dimensional search space. Similarly, the local BO method LA-MCTS also fails due to its high computation cost caused by the SVM boundary for domain decomposition. In contrast, the proposed VGT algorithm continues to lead ahead of other baselines, and the margin with others is much larger in the high-dimensional cases. This demonstrates that the VGT algorithm exhibits much higher sampling efficiency for high-dimensional problems.

Fig. 11 displays a scatter plot of observations, using Ackley200D as an example. The distribution of observations in the local BO methods TuRBO and MCTD is relatively concentrated throughout the optimization process, limiting the algorithm's global exploration ability in the early stage. Conversely, the distribution of observations of the variable selection method MCTS-VS is overdispersed, indicating its weak local exploitation ability and difficulty in achieving rapid descent in the objective function value. In contrast, the observations of VGT are dispersed in the early stage, providing more exploration of the search space. As the iteration progresses, the observations tend to concentrate during the late stage, facilitating more focused exploitation of the search space.

## C.3 SENSITIVITY ANALYSIS OF HYPER-PARAMETERS

We further investigate the sensitivities of the hyper-parameters of VGT, including $C_p$ for exploration and exploitation balance, the number of neighbors $K$ used for approximating the Voronoi graph and the subspace exploitation ratio $R_p$. The corresponding experimental results are visualized in Fig. 12.

**Exploration & exploitation balance parameter** $C_p$   The hyper-parameter $C_p$ balances the exploitation in the best cell and the exploration of sparse areas with fewer visits. A large value of $C_p$ prioritizes exploration over exploitation, leading

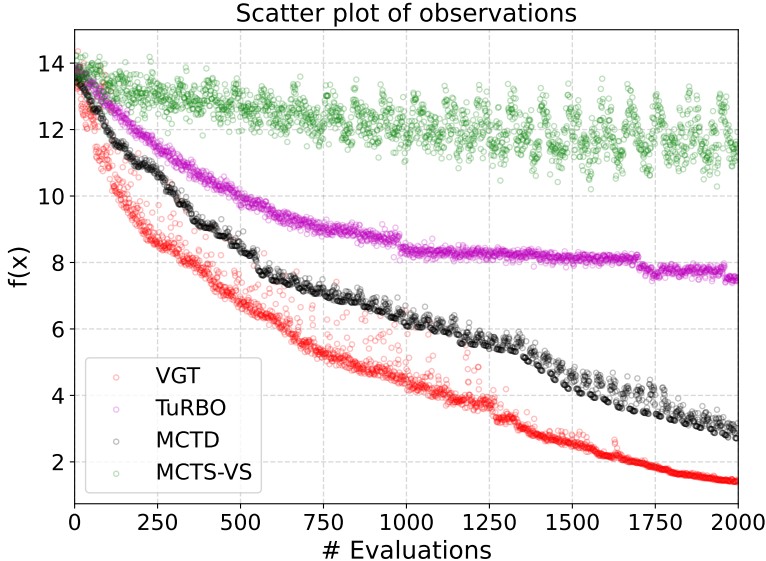

Figure 11: Scatter plot of observations, using Ackley200D as an example.

to the algorithm traversing under-explored regions of the search space. This can result in a reduced convergence rate, as shown in Fig. 12(a) for $C_p = 1$ and $C_p = 5$. Setting $C_p$ within a reasonable range generally does not significantly impact the optimization ability of the algorithm.

**Number of neighbors $K$**  The hyper-parameter $K$ determines the number of nearest neighbors used to approximate the Voronoi graph, which directly affects the performance and computational complexity of the algorithm. A small value of $K$ can lead to a poorly fitted VNGP model, which is not instructive for the optimization process, e.g. $K = 20$ in Fig. 12(b). Meanwhile, a larger value of $K$ improves accuracy at the cost of increased computational burden. Therefore, selecting an appropriate value of $K$ according to the problem size is important to achieve a balance between performance and computational efficiency. For example, the choice $K = 80$ for the 200D problem in Fig. 12(b) yields satisfactory results. However, if $K$ is further increased, the improvement in sampling efficiency is not significant.

**Subspace exploitation ratio $R_p$**  The parameter $R_p$ controls the ratio to perform subspace BO sampling. It is essential to select appropriate $R_p$ according to the intrinsic dimension of the problem. A larger value of $R_p$ can lead to a degradation of the dimension "diversity" of the observations, which can result in the optimization trapped in the sub-optimal region. In our experiments, we observe that just a small $R_p$ can effectively extract the EDR subspace and accelerate the optimization process for problems with redundant dimensions. For instance, setting $R_p = 1/3$ is sufficient for the Ackley10_500D problem in Fig. 12(c).

## C.4 EXPERIMENTAL SETTINGS

We use the opensource implementation of the baselines referred to by the authors: MCTD[1], MCTS-VS[2], TuRBO[3], LA-MCTS[4] and HesBO[5]. For CMA-ES, we use the `pycma` library[6], and for Nelder Mead, we use the Python implementation [7]. We adopt the default hyper-parameter settings by the authors. The detailed experimental configuration is as follows:

**MCTD**  We use the author's default parameter settings with $C_d = 10$ for the weight of recent improvement, $C_p = 0.5$ for the weight of exploration, $C'_p = 0.1$ for branch exploration and $C''_d = 50, C''_p = 0.1$ for leaf exploration.

[1]`https://github.com/yazhai/mctd`
[2]`https://github.com/lamda-bbo/MCTS-VS`
[3]`https://github.com/uber-research/TuRBO`
[4]`https://github.com/facebookresearch/LaMCTS`
[5]`https://github.com/aminnayebi/HesBO`
[6]`https://github.com/CMA-ES/pycma`
[7]`https://github.com/fchollet/nelder-mead`

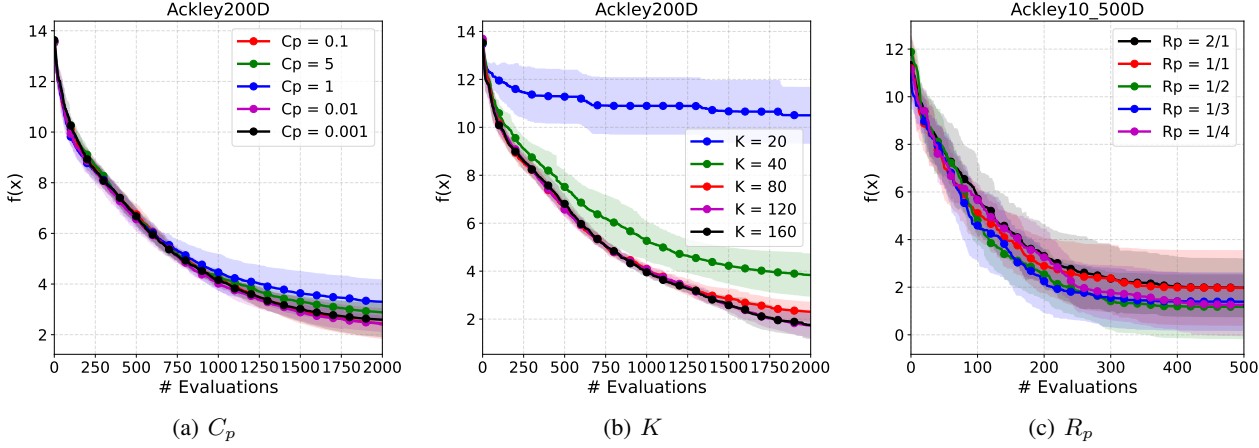

Figure 12: Sensitivity analysis of hyper-parameters.

**TuRBO**    To achieve better performance, we consider a single trust region for TuRBO. The batch size is set to 50 for the 1000D benchmarks and 20 for other benchmarks.

**MCTS-VS**    We use the author's default parameter settings with $k = 20$, $C_p = 0.1$, feature batch size $N_v = 2$ and sample batch size $N_s = 3$.

**LA-MCTS**    We use the parameter settings leaf size=20, $C_p = 0.1$ and gamma type="auto". For the Ackley, Rastrigin and SVM388D benchmarks, the poly kernel is used for the boundary. For other benchmarks, RBF kernel is used. For LaMCTS-TuRBO, we use 20 initial points and a total of 50 evaluations for each TuRBO iteration.

**HesBO**    We set the low dimension $d = 20$ and use the box size $[-0.5, 0.5]^d$ to reduce blind exploration.

**CMA-ES**    We run CMA-ES with $\sigma = 0.1$ and the default population size $p = 4 + \lfloor 3 \cdot \log D \rfloor$.

**Nelder Mead**    We use the parameter settings with $\alpha = 4, \gamma = 8, \rho = 0.1$ and $\sigma = 0.1$.

The local BO of VGT is implemented with `GPyTorch`[8] library and EI is used as the acquisition function. We use the synthetic functions from the SFU benchmarks[9], IEEE analog benchmark circuits[10], vehical design problem Mopta_08 and SVM training task from [11]. The detailed experimental setups and hyper-parameters of VGT are summarized in Table 2.

---

[8]`https://github.com/cornellius-gp/gpytorch`
[9]`https://www.sfu.ca/~ssurjano/optimization.html`
[10]`https://sagroups.ieee.org/2427/analogue-benchmark-circuits/`
[11]`https://arxiv.org/pdf/2103.00349.pdf`

Table 2: Summary of experimental settings

| Benchmarks | Dimension($D$) | Search space | Initial points | Iteration ($T$) | $K$ | $C_p$ | $R_p$ |
|---|---|---|---|---|---|---|---|
| Ackley5D | 5 | $[-5, 10]^5$ | 10 | 60 | 20 | 0.1 | − |
| Griewank5D | 5 | $[-500, 500]^5$ | 10 | 60 | 20 | 0.1 | − |
| Rosenbrock5D | 5 | $[-2.048, 2.048]^5$ | 10 | 60 | 20 | 2 | − |
| Hartmann6D | 6 | $[0, 1]^6$ | 10 | 60 | 20 | 0.1 | − |
| Ackley100D | 100 | $[-5, 10]^{100}$ | 50 | 2000 | 100 | 0.1 | − |
| Ackley200D | 200 | $[-5, 10]^{200}$ | 50 | 2000 | 160 | 0.1 | − |
| Ackley1000D | 1000 | $[-5, 10]^{1000}$ | 50 | 5000 | 300 | 0.1 | − |
| Griewank100D | 100 | $[-500, 500]^{100}$ | 50 | 2000 | 100 | 0.01 | − |
| Griewank200D | 200 | $[-500, 500]^{200}$ | 50 | 2000 | 160 | 0.01 | − |
| Griewank1000D | 1000 | $[-500, 500]^{1000}$ | 50 | 5000 | 300 | 0.01 | − |
| Rosenbrock100D | 100 | $[-2.048, 2.048]^{100}$ | 50 | 2000 | 100 | 2 | − |
| Rosenbrock200D | 200 | $[-2.048, 2.048]^{200}$ | 50 | 2000 | 160 | 2 | − |
| Rosenbrock1000D | 1000 | $[-2.048, 2.048]^{1000}$ | 50 | 5000 | 300 | 2 | − |
| Rastrigin100D | 100 | $[-5.12, 5.12]^{100}$ | 50 | 2000 | 100 | 5 | − |
| Rastrigin200D | 200 | $[-5.12, 5.12]^{200}$ | 50 | 2000 | 160 | 10 | − |
| Rastrigin1000D | 1000 | $[-5.12, 5.12]^{1000}$ | 50 | 5000 | 300 | 10 | − |
| Hartmann6_500D | 500 | $[0, 1]^{500}$ | 10 | 1000 | 30 | 0.1 | 1/1 |
| Ackley10_500D | 500 | $[-5, 10]^{500}$ | 10 | 1000 | 40 | 0.1 | 1/1 |
| OPAMP36D | 36 | $[0, 1]^{36}$ | 10 | 500 | 60 | 0.1 | 1/4 |
| PLL77D | 77 | $[0, 1]^{77}$ | 10 | 500 | 40 | 0.1 | 1/4 |
| Mopta08_124D | 124 | $[0, 1]^{124}$ | 10 | 1000 | 80 | 0.05 | 1/4 |
| SVM388D | 388 | $[0, 1]^{388}$ | 10 | 2000 | 120 | 0.001 | 1/4 |