# OpenReview forum: "Exploring High-dimensional Search Space via Voronoi Graph Traversing"
_auai.org/UAI/2024/Conference — UAI 2024 poster_

### Official Review · Reviewer_fwZ1 · 2024-03-07

**Q2-1 Originality-Novelty:** 3
**Q2-2 Correctness-Technical Quality:** 3
**Q2-5 Clarity Of Writing:** 4

**Q1 Summary And Contributions:**

The article considers the problem of Bayesian optimization (BO) in high-dimensional spaces. The proposed solution divides the search space into Voronoi cells that correspond to the observations. The next explored cell is chosen via UCB-based strategy from the neighbors of the current cell, and the observation within this cell is chosen via the local Bayesian optimization. The experimental results demonstrate that the proposed method is more sample-efficient than the earlier BO methods in the high-dimensional regime.

**Q2-3 Extent To Which Claims Are Supported By Evidence:**

4: Excellent: all claims are supported by very convincing evidence (in the form of comprehensive experimental evaluation, rigorous mathematical proofs, detailed (pseudo-)code, precise references, well-motivated and realistic assumptions) and the authors deliver what they promise.

**Q2-4 Reproducibility:**

3: Good: key resources (e.g. proofs, code, data) are available and key details (e.g. proofs, experimental setup) are sufficiently well-described for competent researchers to confidently reproduce the main results.

**Q3 Main Strengths:**

As far as I know, the main idea of the article (localizing the high-dimensional BO by dividing the space into Voronoi cells that correspond to the current observations) is novel, and seems to improve the results in practice. Also the problem of high-dimensional BO is important, since the basic BO methods do not usually work well in the high-dimensional cases. The article discusses the related work in detail, and the explains how the proposed method differs from the earlier localized BO approaches (the local cells are formed via the Voronoi diagram, as opposed to choosing them as hyperrectangles or learning them via SVM). The article is clearly written, and the approach can be understood from the description. The visualizations are illustrating.

**Q4 Main Weakness:**

The writing seems a little bit careless: there are typos left in the article, and the citation technique is strange (see Q5).

**Q5 Detailed Comments To The Authors:**

The names of the authors should be placed inside the brackets when they are not used as a part of the sentence. For instance, "as discussed by Snoek et al. [2012]" is correct, but "another method, TuRBO Eriksson et al. [2019]", should be "another method, TuRBO [Eriksson et al., 2019]".

**Q9 Complying With Reviewing Instructions:**

Yes

---

> ### Author Rebuttal · Authors · 2024-04-05
>
> Thank you for your valuable feedback. We have carefully reviewed the manuscript to detect and rectify any possible spelling and grammar errors. Furthermore, we have revised the citation format from “TuRBO Eriksson et al. [2019]” to “TuRBO [Eriksson et al., 2019]” for accurate citation formatting.

---

### Official Review · Reviewer_e6Zy · 2024-03-21

**Q2-1 Originality-Novelty:** 3
**Q2-2 Correctness-Technical Quality:** 3
**Q2-5 Clarity Of Writing:** 4

**Q1 Summary And Contributions:**

This paper deals with the problem of optimizing costly black-box functions by means of Bayesian Optimization (BO) methods. However, the application of BO methods is challenging in the high dimensional setting, and an important area of research concerns the efficient exploration of the search space. This paper introduces a Voronoi Graph Traversing (VGT) algorithm that decomposes the search space into Voronoi cells and implements the search problem into a VGT problem. Also a low-dimensional alternative is given. The performance of the algorithm is shown empirically on a number of high-dimensional benchmarks and on a real world problem.

**Q2-3 Extent To Which Claims Are Supported By Evidence:**

2: Fair: the main claims are somewhat supported by evidence (but the experimental evaluation may be weak, or does not match entirely with the claims, important baselines may be missing, proofs contain important ideas but lack rigor, algorithmic details are only discussed superficially, references are imprecise, assumptions are not sufficiently motivated or explicated, etc.).

**Q2-4 Reproducibility:**

3: Good: key resources (e.g. proofs, code, data) are available and key details (e.g. proofs, experimental setup) are sufficiently well-described for competent researchers to confidently reproduce the main results.

**Q3 Main Strengths:**

I think that this is a well-written paper with a clear focus. The algorithm is clearly described and the empirical evaluation convincing.

**Q4 Main Weakness:**

The behaviour of the algorithm is shown empirically, with no formal proof. Indeed, some of the implemented strategies are motivated by the intuition on the behaviour of the objective function. For example, on page 4 it is claimed that "The Voronoi boundaries, ..., are deemed most informative and worthy of exploration". I wonder whether it would be possible to identify some classes of functions where this claim can be shown to be true. Another instance can be found in Section 5.2 where it is claimed that "By combining graph traversal with promising cell sampling, VGT guides towards global optimum...": is that really true? Cannot the algorithm be stuck on a local optimum?

**Q5 Detailed Comments To The Authors:**

This paper deals with the problem of optimizing costly black-box functions by means of Bayesian optimization (BO) methods. However, the application of BO methods is challenging in the high dimensional setting, and an important area of research concerns the efficient exploration of the search space. This paper introduces a Voronoi Graph Traversing (VGT) algorithm that decomposes the search space into Voronoi cells and implements the search problem into a VGT problem. Also a low-dimensional alternative is given. The performance of the algorithm is shown empirically on a number of high-dimensional benchmarks and on a real world problem.

I think that this is a well-written paper with a clear focus. The algorithm is clearly described and the empirical evaluation convincing.

The satisfying behaviour of the algorithm is shown empirically, with no formal proof. Indeed, some of the implemented strategies are motivated by the intuition on the behaviour of the objective function. For example, on page 4 it is claimed that "The Voronoi boundaries, ..., are deemed most informative and worthy of exploration". I wonder whether it would be possible to identify some classes of functions where this claim can be shown to be true. Another instance can be found in Section 5.2 where it is claimed that "By combining graph traversal with promising cell sampling, VGT guides towards global optimum...": is this really true? Cannot the algorithm be stuck on a local optimum?

Please note that the link to github on page 2 is not properly given:
https://github.com/XXX/XXXXXX

**Q9 Complying With Reviewing Instructions:**

Yes

---

> ### Author Rebuttal · Authors · 2024-04-05
>
> Thanks for your valuable comments. I’ll answer your concerns and questions in order.
>
> For Q1, we will elucidate the expressions. The Voronoi boundaries are the perpendicular bisectors between observations. The candidates near boundaries exhibit a larger posterior standard deviation of the GP model compared to areas close to existing observations. Selecting candidates with the maximal posterior standard deviation for the next query maximizes the information gain about the existing observations $X_t$, which is defined as the entropy reduction when knowing the function value $y$ at ${x}$: $I({x}) = H(y)-H(y|X_t)$. This strategy is also known as the pure exploration (PE) acquisition function, which is defined as $PE({x})=\sigma({x})$ [Contal et al., 2013].
>
> For another expression, in the high-dimensional space, the sparse observations make it not easy to be trapped into local optimum. The number of nearby observations needed to confine the optimization process within a local optimum grows exponentially with the problem dimension. This explains why trust region-based local methods, such as TuRBO [Eriksson et al., 2019] and BOBYQA [M. Powell, 2009], perform well for high-dimensional problems. Additionally, the Voronoi graph tends to be a complete graph in high-dimensional space, which makes it easy to traverse to a neighboring cell with a small $h$ through UCB. Moreover, the escape or restart strategy is crucial for local BO methods. If there is no reduction in the objective function value over multiple continuous iterations, we move to the Voronoi cell with minimal $h$ and create a new search path. While we employ these strategies in our implementation to minimize the risk of getting stuck in a local optimum, there remains the potential to encounter such scenarios for some heterogeneous objective functions.
> We have modified the statements in Section 4.1 and Section 5.2 to include these explanations.
>
>
> For Q2, the source code is available as a zipped file in the supplementary material. The link has been temporarily concealed in the manuscript for blind review. The GitHub repository is located at https://github.com/Anonyuser01/VGT .

---

### Official Review · Reviewer_9Rb4 · 2024-03-22

**Q2-1 Originality-Novelty:** 2
**Q2-2 Correctness-Technical Quality:** 1
**Q2-5 Clarity Of Writing:** 3

**Q1 Summary And Contributions:**

The article propose to organise the navigation of high-dimensional space to guide Bayesian Optimization via the use of Voronoi-based structures.
The results are measure in the value of the objective function to minimize against the number of oracle calls.

**Q2-3 Extent To Which Claims Are Supported By Evidence:**

1: Poor: the authors fail to convincingly backup their main claims (e.g., if the experimental evaluation is flawed, proofs are lacking or invalid, references are missing, assumptions are not realistic, not specified, or not motivated).

**Q2-4 Reproducibility:**

1: Poor: key details (e.g. proof sketches, experimental setup) are incomplete/unclear, or key resources (e.g. proofs, code, data) are unavailable.

**Q3 Main Strengths:**

The paper addresses the important topic of enhancing BO in HD spaces.

**Q4 Main Weakness:**

The major weakness, which I consider as a flaw in the analysis, is the argument that using Voronoi-like structures will help resolve the issues in high dimensions (HD).
The first basic points are to resolve the unclear statements of the paper:
* (minor) Voronoi Diagrams are 2D structures. The correct d-dim terminology (d>2) is Dirichlet region
* (important) The difference between what is called a Voronoi Graph (VG) and the multi-dimensional extension of the definition of the Delaunay Graph is unclear. I cannot see any novetly and reason why to change the terminology
* (major) By construction, the VG will suffer the dimensionality as much as any other search structure. In particular, the degree of each vertex will follow the Kissing Number (essentially exponential) due to the sphere packing approximation of the containing empty spheres. The exponential growth of the degree will not only make the use of the graph infeasible but even its storage will somewhat tend to the complete graph (essentially due to the concentration of distance -and angles- phenomenon). As a result, I do not see any supporting argument to use such a structure to resolve issues in HD.
* (major) in fine, the use of the VG boils down to a kNN graph construction, which poses the question of k (again the exponential growth of required NN) and the efficacy of the technique, which could then extend to LSH and go the full circle of kNN search again (see the series of works of Piotr Indyk for example) or use HNSW (Malkov et al, 2014). Here again, I do not see any contribution justifying acceptance.

**Q5 Detailed Comments To The Authors:**

See weaknesses:
* why reinventing terminology when the field of Computational Geometry is quite established?
* how does the proposal resolve the issues in HD space including bypassing the problem of HD-nearest neighbor search?

=> It would be important to analyse the intrinsic dimensionality of the data you provide results on since I do not see in the proposal what makes it better for high intrinsic dimensionality spaces.

**Q9 Complying With Reviewing Instructions:**

Yes

---

> ### Author Rebuttal · Authors · 2024-04-05
>
> We appreciate your comments, but I have to clarify that our aim in this paper is to address the challenge of optimizing expensive black-box functions as described in Eq. (1). These functions entail costly evaluations, typically allowing only several hundred or at most a few thousand observations. The observations are sparsely distributed in HD space. This scenario markedly differs from the dense datasets encountered in k nearest neighbor search problems such as LSH [Indyk et al., 2015] and HNSW [Malkov et al., 2014], which involve millions of points. Bayesian optimization stands out as a sample-efficient approach aimed at minimizing the function evaluations. Specifically, in this paper we concentrate on enhancing the sample efficiency of BO in high-dimensional space by generating the next observation within the promising Voronoi cell. This problem is distinct from solving a k-NNS problem with large datasets.
> Then I’ll answer your concerns and questions in order.
>
> For Q1, the concept of Voronoi diagram is more commonly used than Dirichlet tessellation although even in D-dimensional cases, as exemplified by VOO [Kim et al., 2020], VPS [Rushdi et al., 2016].
>
> For Q2, the graph $G(V,E)$ is the Delaunay graph in our low dimensional implementation of VGT. However, in HD space, the computational cost of establishing an explicit Delaunay graph is prohibitive. Instead, in HD space, we connect observations based on similarity or distance to create a connected graph, where each vertex represents a Voronoi cell for further exploration. The graph is not exactly the Delaunay graph. Therefore, employing the term “Delaunay graph” might lead to misunderstanding for readers. Moreover, the high sample efficiency of our proposed HDBO method primarily stems from confining local Bayesian optimization within promising Voronoi cells, as discussed in Section 4.1. Using Voronoi provides a direct impression of the proposed method. We have revised our manuscript in Section 3 to provide a clear explanation.
>
> For Q3, as you previously mentioned, computing the explicit boundary of Voronoi cells in HD space is impractical. In the proposed HDBO method, our main concern is to ensure that the next observation lies within the target Voronoi cell. As elaborated in Section 4.2, we employ reject sampling to discard candidates located outside the target cell, a strategy akin to that used in VOO [Kim et al., 2020].
>
> For Q4, thanks for your recommendation of the efficient k-NNS approaches.
> LSH [Indyk et al., 2015] and HNSW [Malkov et al., 2014] are computationally efficient k-NNS methods for large datasets with millions of points in HD space. However, in the scenario of BO, typically only several hundred to a few thousand observations are permitted due to the expensive function evaluations. As depicted in Section 6, our experiments involve datasets with 500-5000 observations, which are relatively small and could even be solved by brute force search.
> Moreover, the k-NNS is a component of the proposed HDBO method, but not the primary bottleneck in terms of time. The following results summarize the average time consumption of various components for the HDBO method, using Ackley100D as an example with a total of 2000 iterations. The time consumption of k-NNS is negligible. The main time consumption in HDBO stems from the BO components: training GP model (with a complexity of $O(n^3)$) and optimizing the acquisition function (with the GP prediction complexity of $O(n^2)$). $n$ is the number of observations for training GP. We appreciate your recommendation of efficient k-NNS methods and will consider using them to address higher-dimensional problems that require larger sample budgets of $\sim 10^6$. We have revised our manuscript in Section 4.3, Section 5.1 and Appendix C.1 to include these discussions.
>
> Algo. Phases | k-NNS | Train GP model | Optimize acquisition function | Total time consumption
>
> Runtime(s) | $0.80\pm 0.03$ | $1724.91\pm 80.14$ | $855.71 \pm 49.29$ | $2581.42 \pm 99.40$
>
> For the final question regarding dataset intrinsic dimensionality, observations from the BO procedure match the problem dimension. Due to the nature attributes of BO, the observed dataset involves at most a few thousand observations, which are very sparsely distributed in HD space. Especially in early iterations of BO, only a few dozen observations are involved. This contrasts starkly with the k-NNS problem in LSH [Indyk et al., 2015] and HNSW [Malkov et al., 2014], which handle large datasets with millions of points.

---

### Official Review · Reviewer_bxPR · 2024-03-23

**Q2-1 Originality-Novelty:** 2
**Q2-2 Correctness-Technical Quality:** 1
**Q2-5 Clarity Of Writing:** 2

**Q1 Summary And Contributions:**

This submission concerns the black-box function optimization problem, where one wants to compute $x^*=arg \min_{x \in \mathcal{X}} f(x)$ for some function $f$ given by a black box. The problem has been previously tackled using three techniques: dimension decoupling, sub space embedding, and region restriction. This submission proposes a Voronoi Graph Traversing - based algorithm (VGT) which employs Voronoi diagrams to segment the search space. The authors show how to determine the search direction and step length, that gives an expansion and propagation based framework that favors promising cells over ones where the function value is likely to be  sub-optimal.

The authors perform experiments on low-dimensional synthetic benchmarks, high-dimensional synthetic benchmarks, and two real-world optimization problems.The focus is on the high-dimensional setting, and the experiments indicate that the VGT method requires fewer evaluations of $f$ compared to existing methods such as MCTD, TuRBO and La-MCTS, MCTS-VS, HesBO, CMA-ES, the simplex method, and random search.

**Q2-3 Extent To Which Claims Are Supported By Evidence:**

3: Good: the main claims are supported by convincing evidence (in the form of adequate experimental evaluation, proofs, (pseudo-)code, references, assumptions).

**Q2-4 Reproducibility:**

3: Good: key resources (e.g. proofs, code, data) are available and key details (e.g. proofs, experimental setup) are sufficiently well-described for competent researchers to confidently reproduce the main results.

**Q3 Main Strengths:**

The main strength of this work is in the experiments performed to evaluate the method. I found the experiments thorough, with a fair comparison to many existing methods, and sufficient datasets.  In addition, I think the idea of using local BO on neighboring Voronoi cells is nice.

**Q4 Main Weakness:**

Major: I believe the word “Voronoi” in the title and the name of the method is misleading.

The paper starts with a thorough literature review, and the main drawback of existing methods is cited as their scalability to high dimensions.

It is well-known that high-dimensions and Voronoi diagrams do not go hand in hand. I read with interest Sections 3 and 4, where the authors describe the VGT method, and the explanation is in terms of the Voronoi diagram. Obviously I was curious as to how the authors were computing or approximating Voronoi cell information.

However, it is only in Section 4.3 when the reader finds out that the method is actually based on the $K$ nearest neighbor graph, and not the Voronoi graph. In my opinion, this makes the theory presented in Sections 3 and 4 moot, as the implemented method uses nothing from the Voronoi diagram or its properties. All of the exposition could simply have been in terms of the $K$ nearest neighbor graph.

**Q5 Detailed Comments To The Authors:**

Major: Please explain why the exposition regarding Voronoi diagrams is necessary to explain the implemented method. If it is used only for the purpose of providing intuition, it should be clearly stated.

Minor: In Section 5.1, why is the NNS complexity not dependent on the number of points in the dataset or the dimension, and only $O(t)$?

There are several typos (e.g., page 2, “ we aim to sacle..”), and I strongly recommend a spell check. “Newly” should be changed to “new” in several places. Also, I would recommend adding a “by” between the name of the method, and the authors for that method. In my opinion sentences containing “ADD-GP Kandasamy et al”, “TuRBO Ericsson et al” read a bit weird.

**Q9 Complying With Reviewing Instructions:**

Yes

---

> ### Author Rebuttal · Authors · 2024-04-05
>
> Thanks for your valuable comments. I’ll answer your concerns and questions in order.
>
> The Voronoi diagram serves not only as an intuitive concept but also as an essential core of the proposed HDBO method. The implemented method strongly relies on the basics of Voronoi diagram.
> 1) The Voronoi cell restricts the region for optimizing the acquisition function (AF) to create the next query point. We use a Gaussian distribution to sample the AF within the selected Voronoi cell. With the basic property of Eq. (2), the samples lying outside the target cell are discarded by reject sampling. Just like other local BO methods TuRBO [Eriksson et al., 2019] and LA-MCTS [Wang et al., 2020], the profile of Voronoi cell serves as the most important factor that impacts the sample efficiency of HDBO. Explicit computation of Voronoi boundaries is intractable and unworthy in both storage and computation in HD space. To avoid the costly computation of Voronoi boundaries, we use the similar reject sampling strategy as VOO (Voronoi optimistic optimization) [Kim et al., 2020] and LA-MCTS[Wang et al., 2020] to restrict the optimization of AF within the target Voronoi cell. This part is the same in our implementation of VGT in both HD and low dimensional (LD) cases.
> 2) The Voronoi neighbors are used to train the local GP model for the LD case, as discussed in Section 4.1-4.2. The LD algorithm is implemented in our provided source code and presented in the results in Fig. 5 (labeled as “VGT-lowdim”). However, in HD space, explicitly computing the Voronoi neighbors is impractical, thus we train local GP with the K nearest neighbors. For a larger K, the Voronoi neighbors with be included in the K nearest neighbors. For a moderate K, the Voronoi neighbors and the K nearest neighbors mostly tend to be the same.
> Additionally, GP is a statistical model characterized by large uncertainty, and it introduces diagonal noise $\sigma_n^2 I$ to ensure the numerical stability. Therefore, setting a reasonable value for K, training the local GP with K nearest neighbors has little impact on the algorithm’s performance. The ablation experiments in Fig. 5 show that the VGT method with exact Voronoi neighbored GP (labeled as “VGT-lowdim”) and K nearest neighbored GP (labeled as “VGT”) exhibit comparable performance, better than other BO baselines.
>
> The high sample efficiency of the proposed HDBO method primarily arises from restricting local BO within the promising Voronoi cell. Therefore, the Voronoi diagram is not merely an intuition but a key factor that ensures the sample efficiency. We have modified the manuscript in Section 4.3 and Section 6.1.1 to add corresponding explanations, so that readers can understand the algorithm more clearly.
>
>
> For Q2, thanks for your suggestion. The NNS implemented by k-d tree has a construction complexity of $\mathcal{O}(D*N\log N)$, an average query complexity of $\mathcal{O}(\log N)$, and a worst-case query complexity of $\mathcal{O}(N)$. The construction complexity depends on the dimension $D$, while the query complexity only corresponds to the number of observations $N$. We've revised our manuscript to incorporate these clarifications.
>
> For Q3, thanks for your correction. We have thoroughly reviewed the manuscript to identify and rectify any potential spelling and grammar errors. Additionally, we have revised the citation format from “ADD-GP Kandasamy et al. [2015]” to “ADD-GP [Kandasamy et al., 2015]” for better readability.

---

### Official Review · Reviewer_edk1 · 2024-03-23

**Q2-1 Originality-Novelty:** 3
**Q2-2 Correctness-Technical Quality:** 3
**Q2-5 Clarity Of Writing:** 4

**Q1 Summary And Contributions:**

The paper proposes a new Bayesian optimization method called VGT for high-dimensional problems. The method employs graph traversal on an (approximate) Voronoi graph using UCB, and performs local Bayesian optimization within selected cells. The proposed method achieves state-of-the-art performance on several standard high-dimensional data sets when compared to recent methods for high-dimensional Bayesian optimisation.

**Q2-3 Extent To Which Claims Are Supported By Evidence:**

3: Good: the main claims are supported by convincing evidence (in the form of adequate experimental evaluation, proofs, (pseudo-)code, references, assumptions).

**Q2-4 Reproducibility:**

4: Excellent: key resources (e.g. proofs, code, data) are available and key details (e.g. proof sketches, experimental setup) are comprehensively described for competent researchers to confidently and easily reproduce the main results.

**Q3 Main Strengths:**

- The method scales to ultra-high dimensional problems and achieves excellent performance in the authors' experiments.
- The method is compared to other recent methods including MCTD, MCTS-VS and LA-MCTS.
- The paper is well-written and presents the method in detail.
- The authors provide a Python implementation of their method.

**Q4 Main Weakness:**

- The method is somewhat ad-hoc, and no ablation studies are performed for the various approximations (Voronoi approximation, subspace BO sampling).
- The authors mention that VGT exhibits lower computational complexity than LA-MCTS, but do not perform any experiments measuring wall-clock time.
- The code to reproduce the experiments is not available.

**Q5 Detailed Comments To The Authors:**

- Can the authors provide a possible explanation on the poor performance of LA-MCTS in the experiments as it is barely better than random search? For example, on Ackley-100d, LA-MCTS is much worse than in the experiments of the original LA-MCTS paper (see Appendix A.2).
- Can the authors clarify whether they fix (most of) the hyperparameters for the compared methods (Appendix C.2) and vary hyperparameters for their own method (Table 1 in Appendix)? This would seem unfair as methods can be quite sensitive with respect to some hyperparameters (e.g. setting K badly can have a significant effect on VGT).
- O(log n) average time complexity for k-d tree is misleading as it is well-known that in even moderate dimensions (d >~ 10), brute force search is more efficient (I would encourage the authors to evaluate this). Moreover, the presented time complexities should be written as dependant on the dimension d.
- I think the VGT pseudocode (Algorithm 1 in Appendix A.2) is important enough that it should be presented in the main paper.

**Q9 Complying With Reviewing Instructions:**

Yes

---

> ### Author Rebuttal · Authors · 2024-04-05
>
> Thank you for your valuable comments. I’ll answer your concerns and questions in order.
> The results of ablation experiments on Voronoi approximation and subspace BO sampling are respectively incorporated in Fig. 5 and Fig. 7. In Fig. 5, the "VGT-lowdim" (without Voronoi approximation) and "VGT" (with Voronoi approximation) exhibit comparable performance. In the results shown in Fig. 7, "VGT-subspace" with subspace BO sampling outperforms "VGT" for the intrinsic low-dimensional problems.
> Regarding the wall-clock time, we will summarize the average runtime of various methods in Appendix C.1.
> We provide the source code as a zipped file in the supplementary material. The link is temporarily concealed in the manuscript for blind review. The GitHub repository is available at https://github.com/Anonyuser01/VGT .
>
>
> For Q1, the experiments in Appendix A.2 of LA-MCTS [Wang et al., 2020] utilize LA-MCTS to enhance TuRBO, resulting in similar performance to TuRBO. In our experiments, we employ LA-MCTS to enhance BO (GP-EI). Due to BO’s poor performance in high-dimensional (HD) space, the improvement by LA-MCTS is limited. Moreover, LA-MCTS’s performance heavily relies on the compatibility between the SVM kernel and the objective function landscape. When the objective function observations are well-separated by the SVM kernel, it guides optimization effectively and obtains good results. Conversely, the SVM boundary will overfit due to sparse observations in HD space and mislead the optimization process. We provide results using Ackley100D as an example, with a total of 500 evaluations comparing the rbf kernel and poly kernel as follows. The rbf kernel aligns better with Ackley100D, achieves better convergence but with much higher computational cost. For practical black-box optimization problems, it is difficult to select the matched kernel due to the unknown function landscape. We will include the results of GP-EI in Fig. 6 and Fig. 10 to explain this.
>
> Methods | BO(GP-EI) | LA-MCTS with poly kernel | LA-MCTS with rbf kernel | VGT
>
> f(x)    | $12.77\pm 0.56$ | $12.61 \pm 0.28$ | $8.62 \pm 0.19$ | $4.25 \pm 0.29$
>
> Runtime | $1.25 \pm 0.04$ h | $20.64\pm 3.76$ min | $11.84 \pm 3.19$ h | $8.62 \pm 0.11$ min
>
> For Q2, our primary purpose is to set a reasonable value for K according to given dimension scales. The performance of VGT is insensitive to other hyperparameters. Typically, we choose $K \approx 10*\lfloor\sqrt{D} \rfloor$ as the default setting for VGT, where D is the problem dimension.
>
> For Q3, thank you for your suggestions. We have evaluated the performance and runtime of VGT with Ackley100D using both k-d tree and brute force search methods. The results of 10 repeated runs with a total of $N=2000$ evaluations are summarized as follows. Both methods exhibit comparable convergence performance, which surpasses other HDBO baselines. The VGT with brute force search has a longer runtime.
> We evaluated the performance and runtime of VGT with Ackley100D for both using k-d tree and brute force search. The results of 10 repeated runs with a total of $N=2000$ evaluations are summarized as follows. The convergence performance of both is comparable and superior to other BO baselines. The VGT with brute force search exhibits longer runtime.
> The k-d tree has a construction complexity of $\mathcal{O}(D*N\log N)$, an average query complexity of $\mathcal{O}(\log N)$, and a worst-case query complexity of $\mathcal{O}(N)$. We have revised the statement in Section 5.1 and Appendix B.
>
> Methods | VGT with k-d tree | VGT with brute force search
>
> f(x)    | $0.39\pm 0.30$ | $0.53 \pm 0.46$
>
> Runtime | $46.56 \pm 1.48$ min | $69.65\pm 2.73$ min
>
>
> For Q4, thank you for your suggestion. We have revised our manuscript and placed the VGT pseudocode on page 5.

---

### Meta-Review · Area_Chair_6Vrm · 2024-04-18

The paper focuses on Bayesian optimization (BO) in high-dimensional spaces and proposes a geometric approach to direct the search based limited function evaluations.

There are clear differences of opinion among the reviewers which are to some extent unresolved in the discussions.

I find that while there are open issues, e.g., related to the role of the the Voronoi tesselation (which is not used in practice due to its intractability in high dimensions. so as reviewer bxPR points out, it seems to only serve the purpose of providing intuition to a method that actually uses nearest neighbor graphs due to their computational simplicity), technically, the work also makes a solid contribution.

If accepted, the authors are requested to improve on the above issues and others pointed our in the reviews.